# Enhancing Cardiovascular Risk Prediction with a Simplified Carotid IMT Protocol: Evidence from the IMPROVE Study

**DOI:** 10.3390/biomedicines13030584

**Published:** 2025-02-26

**Authors:** Fabrizio Veglia, Anna Maria Malagoni, Mauro Amato, Rona J. Strawbridge, Kai Savonen, Philippe Giral, Antonio Gallo, Matteo Pirro, Bruna Gigante, Per Eriksson, Douwe J. Mulder, Beatrice Frigerio, Daniela Sansaro, Alessio Ravani, Daniela Coggi, Roberta Baetta, Nicolò Capra, Elena Tremoli, Damiano Baldassarre

**Affiliations:** 1Maria Cecilia Hospital, GVM Care & Research, 48033 Cotignola, Italy; fveglia@gvmnet.it (F.V.); amalagoni@gvmnet.it (A.M.M.); etremoli@gvmnet.it (E.T.); 2Centro Cardiologico Monzino IRCCS, 20138 Milan, Italy; beatrice.frigerio@ccfm.it (B.F.); daniela.sansaro@ccfm.it (D.S.); alessio.ravani@ccfm.it (A.R.); daniela.coggi@ccfm.it (D.C.); roberta.baetta@ccfm.it (R.B.); nicolo.capra@gmail.com (N.C.); damiano.baldassarre@ccfm.it (D.B.); 3Department of Medicine Solna, Division of Cardiovascular Medicine, Karolinska Institutet, Stockholm, Karolinska University Hospital, 17177 Solna, Sweden; rona.strawbridge@glasgow.ac.uk (R.J.S.); bruna.gigante@ki.se (B.G.); per.eriksson@ki.se (P.E.); 4School of Health and Wellbeing, University of Glasgow, Glasgow G12 8TB, UK; 5Health Data Research UK, Glasgow G12 8TA, UK; 6Foundation for Research in Health Exercise and Nutrition, Kuopio Research Institute of Exercise Medicine, 70100 Kuopio, Finland; kai.savonen@uef.fi; 7Department of Public Health and Clinical Nutrition, University of Eastern Finland, 70210 Kuopio, Finland; 8INSERM, Unité de Recherche sur les Maladies Cardiovasculaires, le Métabolisme et la Nutrition, ICAN, Sorbonne Université, F-75013 Paris, France; philippe.giral@aphp.fr (P.G.); antonio.gallo@aphp.fr (A.G.); 9Lipidology and Cardiovascular Prevention Unit, Department of Nutrition, APHP, Sorbonne Université, Hôpital Pitié-Salpêtrière, F-75013 Paris, France; 10Internal Medicine, Angiology and Arteriosclerosis Diseases, Department of Medicine and Surgery, University of Perugia, 06129 Perugia, Italy; matteo.pirro@unipg.it; 11Department of Internal Medicine, University Medical Center Groningen, University of Groningen, 9700 RB Groningen, The Netherlands; d.j.mulder@umcg.nl; 12Department of Medical Biotechnology and Translational Medicine, Università degli Studi di Milano, 20129 Milan, Italy

**Keywords:** carotid artery, intima-media thickness, plaque, ultrasonography protocol, IMPROVE study

## Abstract

**Background/Objectives**: Carotid intima-media thickness (CIMT) has long been used as an index of subclinical atherosclerosis, but its role as a risk modifier in cardiovascular (CV) risk optimization has recently been questioned due to methodological problems, such as lack of protocol standardization and scanning difficulties. In this multicentre, longitudinal, and observational study, we tested the predictive ability of two new CIMT variables detectable with a simplified, quick, and easy-to-standardize protocol. **Methods**: CIMT was measured in 3165 subjects from six centers, in five European countries, belonging to the IMPROVE study. The two variables tested were the average of two maximal CIMT measures taken, from a single angle, in the right and left common carotids (1CC-IMT_mean-of-2-max_) or bifurcations (BIF-IMT_mean-of-2-max_). The ability to predict CV events, on top of the SCORE2/SCORE2-OP risk algorithm, was quantified by the time-dependent increase in the receiver operating characteristic (ROC) area under the curve (AUC). **Results**: During a median follow-up of 7.1 years, 367 cardio-, cerebro-, and peripheral-vascular events were registered. Both CIMT variables tested were associated with CV risk, but 1CC-IMT_mean-of-2-max_ was also able to significantly increase the ROC AUC over the risk score (+0.017, *p* = 0.014). The result was stable after running several sensitivity analyses. **Conclusions**: 1CC-IMT_mean-of-2-max_ is able to significantly improve the predictive capacity of SCORE2/SCORE2-OP. Being based on a simple and easily standardized measurement protocol, this new variable is a promising candidate for application in mass screening and risk assessment in primary prevention.

## 1. Introduction

For more than two decades, carotid intima-media thickness (CIMT), assessed by B-mode ultrasonography, has been considered a well-recognized index of subclinical carotid atherosclerosis [1,2,3,4,5], has been widely employed as a surrogate endpoint of cardiovascular (CV) disease in clinical trials [6], and has been proposed as a valuable predictor of future CV events [7,8,9] and as a marker to assess progression and regression of atherosclerosis [10]. Recently, however, these traits have been questioned so that most international guidelines no longer recommend the measurement of CIMT for risk assessment [11]. Among the other imaging techniques employed for CV risk refinement, coronary artery calcium (CAC) measurement has gained popularity, but its employment in primary prevention is recommended only for subjects at intermediate risk [12].

In the past, many groups—including ours—have documented the ability of CIMT to significantly improve prediction over that obtainable with global CV risk calculation algorithms, e.g., the Framingham risk score. Key factors for optimizing the predictive ability of CIMT, which have emerged from previous experience, include (1) the number of carotid segments measured, (2) whether atherosclerotic plaques are included in the measurement process, and (3) the type of measure employed (mean or maximal IMT value).

A recent meta-analysis showed that summary measures combining several CIMT values, obtained in different segments of the carotid arteries, are more strongly associated with CV events than single measurements and that maximal values perform better than mean values [13]. To be noticed, the ultrasonographic variable most widely used in epidemiological studies, and now downgraded in most international guidelines, is based on mean measures usually taken in a single segment. Moreover, many studies employ IMT measures taken only in plaque-free (PF) areas [14,15,16,17] and the 2021 Guidelines [11] maintain a clear distinction between CIMT (poorly predictive and not recommended) and carotid plaques (more predictive and still recommended).

In our experience, the highest level of prediction accuracy was achieved by combining IMT measurements from various carotid segments into a composite variable, called IMT_mean–max_ [7]. On the other hand, despite its better performance, the protocol for IMT_mean–max_ measurement demands well-trained operators and is rather time-consuming, since it requires 20–25 min for a high-quality scan. All this may limit its use for risk stratification in clinical settings or for mass screening. Therefore, there is a need to identify new ultrasonographic variables that are easier to implement, suitable for very large studies, and whose protocol can be easily standardized.

The present study aimed to assess whether two new CIMT composite variables, obtained using less demanding measurement protocols than that required for measuring IMT_mean–max_, still have an adequate predictive ability, comparable to that of IMT_mean–max_ and better than that obtained with mean single segment IMT or with IMT measured in PF areas.

For this purpose, we conducted a reanalysis of the IMPROVE database, a large multicenter, longitudinal and observational study that included over 3000 participants from five countries across southern, western, and northern Europe [7].

## 2. Materials and Methods

### 2.1. Participants and Design

Between 2004 and 2005, the original IMPROVE study enrolled 3711 individuals, aged 54–79 years, with at least three vascular risk factors (VRFs) and no prior cardio- or cerebro-vascular events [18]. Subjects, at medium/high CV risk, were recruited in seven European centers: two in Italy (Milan and Perugia), one in France (Paris), one in the Netherlands (Groningen), one in Sweden (Stockholm), and two in Finland (Kuopio-1 and Kuopio-2). In 2023, the Kuopio-1 center (Institute of Public Health and Clinical Nutrition at the University of Eastern Finland) withdrew permission to use its data. Therefore, the sample used in the present study includes six centers and 3165 subjects (Flow diagram in Appendix A). Candidate VRFs and inclusion and exclusion criteria were previously described [18].

In 2016, the follow-up of subjects recruited in four centers (Milan, Perugia, Paris, and Stockholm) was extended to include the latest available information on life status, causes of death, and incident CV events. Subjects were actively searched by telephone contacts, hospital records, and institutional databases. The two remaining centers (Kuopio-2 and Groningen) did not provide any additional information, so the follow-up of these two centers remained limited to three years.

The IMPROVE study adheres to the standards of good clinical practice and ethical principles of the Declaration of Helsinki and was approved by local ethics committees.

Informed consent was obtained from all subjects involved in the study.

### 2.2. CV Risk Estimation Algorithm

To evaluate the ability of the new CIMT variables to enhance the prediction of CV events beyond existing algorithms, we initially estimated individual risk using SCORE2/SCORE2-OP [19,20]. These models were selected because they are specifically tailored to the European population and account for baseline risk variations across different countries.

### 2.3. Carotid IMT Measurements

Details on the original ultrasound protocol of the IMPROVE study have been previously reported [7]. Briefly, all centers were equipped with identical machines (Technos, Esaote, Genoa, Italy), each featuring a 5 to 10 MHz linear array probe. Calibration was performed using a phantom at baseline and after one year. The far walls of the common carotid arteries, bifurcations, and internal carotid arteries were visualized at three scan angles (lateral, anterior, and posterior) and recorded on sVHS videotapes (later transferred onto DVDs). All measurements were performed at the Milan center using specialized software (M’Ath, Metris SRL, Argenteuil, France, version 2.0 (in use in 2012). Both sonographers and readers were trained and certified by the coordinating center in Milan.

In each of the eight segments, the “mean” and the “max” IMT were recorded. The mean value was the average of all the measures obtained by the same dedicated software in that specific segment, including any plaques present (defined as IMT > 1.5 mm) and PF areas. The max value was the thickest measure detected in the same segment. The original IMT_mean–max_ variable was computed as the average of the local maximal values detected in the following carotid segments: 1st cm of the internal carotid artery (ICA), bifurcation (Bif), 1st cm of the common carotid artery (1CC), and the rest of the common carotid artery, corresponding to the portion from cm 2 to 4 (CC). Both left and right carotid arteries were included, for a total of eight maximal values [7].

For the purposes of the present study, we calculated two further ultrasonographic variables (i.e., 1CC-IMT_mean-of-2-max_ and BIF-IMT_mean-of-2-max_; Figure 1A and Figure 1B, respectively) that are less demanding, in terms of scan time and measurement complexity, than IMT_mean–max_.

First, these variables were derived by considering a single angle (lateral) instead of three, based on the assumption that a single projection provides sufficient information to serve as a proxy of the other two angles. Then, we restricted the area to either the bifurcation (Bif) or the 1st cm of the common carotid (1CC) instead of the whole carotid tree. Bif was selected because it is the area with the highest probability of carrying a plaque (in our sample, 58% of subjects had a maximal IMT value > 1.5 mm in the Bif); 1CC was chosen because it is easier to visualize and measure than Bif, although plaques were less frequent in this segment (12% in our sample). The ICA was not considered because its visualization is more complex, involving longer scan times, increased subjectivity, a greater rate of missing data (in our sample 1.1% vs. 0.7% for Bif and 0.03% for the 1CC), and higher inter-sonographer variability [18].

Our goal was to maintain the “mean–max” concept, that is, to average several maximum values found in different areas of the carotid artery. At the same time, we wanted to limit the number of segments to be visualized so that the operator did not have to scan the entire carotid tree. In order to achieve a reasonable balance between speed and ease of execution on the one hand, and the ability to assess the degree and extent of atherosclerotic burden on the other, each of the new variables was based on a minimum of two measurements, one in the right and one in the left artery. 1CC-IMT_mean-of-2-max_ was computed as the average of the max thickness of the 1st cm of left CC and of the max thickness of the 1st cm of right CC; BIF-IMT_mean-of-2-max_ was computed as the average of the max thickness of the left Bif and of the max thickness of the right Bif. Like in the original IMT_mean–max_, in the new variables local maxima were also assessed regardless of the presence or absence of plaques in the segments considered.

To be noticed, the two new variables differ from other IMT measures widely used in clinical studies, even by our own group. For example, the 1CC-IMT_mean-of-2-max_ differs from 1CC-IMT_max_, a variable we previously considered [7], because the latter is computed as the overall maximal thickness detected in left and right 1st cm of CC; being the measure of one single point in the whole carotid tree, it is less representative of the whole atherosclerotic burden and is potentially more subject to statistical variability.

The prediction ability of 1CC-IMT_mean-of-2-max_ and BIF-IMT_mean-of-2-max_ was compared with that of IMT_mean–max_. A further variable, PF-CC-IMT_mean_ (the average of all measures taken in the plaque-free areas of the common carotids), was also compared, as this is the IMT variable commonly employed in clinical and epidemiological studies. The definitions of these ultrasonographic variables are reported in Appendix A.

In a sensitivity analysis, we also tested the predictive power of the “presence of a carotid plaque” by collapsing the quantitative information provided by CIMT into several categorical dichotomous variables considering different segments (CC, Bif, or whole carotid tree, including ICA).

### 2.4. Statistical Analysis

Baseline numerical variables were summarized as mean ± SD or as median (interquartile range), according to their distribution, and were compared between subjects with or without events via the Wilcoxon rank-sum test. Categorical variables were summarized as frequency (%) and compared via the chi-squared test.

Time-dependent receiver operating characteristics (ROC) curves were used to assess the capacity of ultrasonographic variables to improve prediction on top of an established CV risk score, i.e., the SCORE2/SCORE2-OP. In addition, in a sensitivity analysis, we also tested the predictive capacity of the new CIMT variables on top of the Framingham risk score.

Time-dependent ROC curves were estimated at 10 years follow-up via Harrel’s method using the “concordance” option in the PHREG SAS version 9.4 procedure. This time span was chosen because major CV risk scores (like Framingham risk score and SCORE2/SCORE2-OP itself) are calibrated to compute cumulative 10-year risk. Pairwise comparisons of time-dependent ROC area under the curves (AUCs) were performed using the inverse probability of censoring weighting approach [21].

The association of ultrasonographic variables with CV endpoints was assessed by computing standardized hazard ratios (HRs) in Cox proportional hazard regression models. The resulting HRs correspond to the expected risk increment for a one standard deviation (SD) increase in the predictor. Standardized HRs for combined events, associated with different ultrasonographic variables, were also computed by alternately splitting the sample into two groups according to sex, age (lower and higher than the median, i.e., 64.5 years), geography (northern centers, i.e., Groningen, Stockholm and Kuopio, versus southern centers, i.e., Paris, Milan and Perugia), personal history of diabetes mellitus, and baseline CV risk (SCORE2/SCORE2-OP lower and higher than the median, i.e., 8.25). The interactions between groups and ultrasonographic variables were also computed.

Quintile-specific HRs were also calculated to assess the shape of the relationship between the ultrasonographic variables and the risk of CV events. Deviation from linearity was tested visually from Martingale residuals and by the Kolmogorov-Type Supremum Tests with 1000 resamplings.

Calibration was assessed by stratifying the two proposed variables into deciles and plotting the number of events observed in each stratum vs. those expected according to the logistic models. Departure from linearity was evaluated via the Hosmer–Lemeshow test, a non-significant *p*-value indicating an adequate calibration.

Except when specified, analyses were not adjusted for the recruiting center, because SCORE2/SCORE2-OP already accounts for potential differences due to geography. Nevertheless, to assess potential biases, sensitivity analyses were run by adjusting also for the recruiting center. Reclassification analysis, with computation of the NRI (Net Reclassification Improvement) and the IDI (Integrated Discrimination Improvement), was not employed because it has been repeatedly criticized, mainly for the high number of false positives produced [22,23,24].

Subjects with missing data were excluded from the relevant analyses. All analyses were performed via SAS version 9.4 (SAS Institute Inc., Cary, NC, USA).

## 3. Results

### 3.1. Baseline Characteristics of Subjects

The baseline characteristics of the 3165 subjects included in the analysis are reported in Table 1.

The median within-center follow-up was 3.0 years for Kuopio and Groningen, 8.7 for Milan, 9.1 for Paris, 10.7 for Stockholm, and 18.1 for Perugia. The median follow-up of the whole cohort was 7.1 years (interquartile range: 3.0; 11.0). During such follow-up, a total of 367 CV events were recorded (incidence per 1000 person-years: 14.8, 95% CI: 13.3; 16.3). Of these, 209 were coronary, 128 cerebrovascular, and 30 peripheral vascular events. Figure 2 shows the Kaplan–Meier curves stratified by SCORE2/SCORE2-OP class.

Event types are detailed in Appendix A.

### 3.2. Predictive Ability of Ultrasonographic Variables

Table 2 reports the performance to predict the combined event, expressed as the ROC AUC for each one of the four ultrasonographic variables of interest (1CC-IMT_mean-of-2-max_, BIF-IMT_mean-of-2-max_, IMT_mean–max_, and PF-CC-IMT_mean_) alone, for SCORE2/SCORE2-OP alone, and for the total model including both.

The AUC increase represents the increment of the predictive ability due to the ultrasonographic variable used on top of the SCORE2/SCORE2-OP. The AUC for SCORE2/SCORE2-OP was 0.645 in all analyses and the AUC increase was significant for IMT_mean–max_ and for 1CC-IMT_mean-of-2-max_. Adjustment for recruiting centers resulted in minimal changes, with AUC increases of 0.027 (95% CI 0.007; 0.047), 0.016 (95% CI 0.004; 0.029), and 0.008 (95% CI −0.006; 0.022) for IMT_mean–max_, 1CC-IMT_mean-of-2-max_, and BIF-IMT_mean-of-2-max_, respectively.

When an endpoint consisting of sudden death, acute myocardial infarction (AMI), and stroke (163 total events) was considered, the AUC increases were very similar to those obtained with the combined event (Appendix A). Due to the lower number of events, the confidence limits were wider and, consequently, the significance was lower.

If 1CC-IMT_mean-of-2-max_ and BIF-IMT_mean-of-2-max_ were simultaneously included in the same prediction model for combined events, the total AUC increase, on top of SCORE2/SCORE2-OP, was 0.018 (95% CI 0.001; 0.036), being slightly more than with 1CC-IMT_mean-of-2-max_ alone, but still less than with IMT_mean–max_.

In order to evaluate the relevance of the AUC increase when the ultrasonographic variables were added to an algorithm for global risk score assessment, we tested the impact of adding one of the variables already included in such algorithm to a hypothetical prediction model that included the other risk factors that are part of the same algorithm (age, sex, smoking status, systolic blood pressure (SBP), low-density lipoprotein cholesterol (LDL-C), and high-density lipoprotein cholesterol (HDL-C) if the algorithm is the SCORE2/SCORE2-OP): the observed AUC increase for “age” (1.6%) (Appendix A) was lower than that observed when 1CC-IMT_mean-of-2-max_ was added to SCORE2/SCORE2-OP (1.7%). When the same test was repeated by adding the variables “sex” and “SBP” to a hypothetical prediction model that was lacking them, the observed increases in AUC were 1.2% and 0.2%, respectively (Appendix A).

### 3.3. Association of Ultrasonographic Variables with CV Events

Standardized HRs for the combined event, according to the different ultrasonographic variables, are reported in Table 3.

The analysis was stratified by center and the HRs were adjusted for SCORE2/SCORE2-OP. The standardized HRs of 1CC-IMT_mean-of-2-max_ and BIF-IMT_mean-of-2-max_ were very similar, with an estimated ~20% risk increment for one SD increase in both ultrasonographic variables. In line with previous results [7], the highest HR was observed for IMT_mean–max_ with an estimated 35% risk increment for one SD increase. In contrast with its poor performance in the ROC AUC analysis, the standardized HR of PF-CC-IMT_mean_ was rather high, showing a 27% increase in risk for each SD increment.

The shape of the relation between the different ultrasonographic variables and the risk of events is depicted in Appendix A (quintiles boundaries of ultrasonographic variables are shown in Appendix A). A fairly linear relationship was observed between the mean value of the quintiles and the HRs of the combined event. The departure from linearity was significant for none of the ultrasonographic variables, with *p*-values ranging from 0.16 to 0.72.

### 3.4. Subgroup Analysis

Appendix A shows the standardized HRs for combined events, associated with different ultrasonographic variables, computed by alternately splitting the sample into two groups according to sex, age, geography, personal history of diabetes mellitus, and baseline CV risk. In general, the association of IMT_mean–max_ and 1CC-IMT_mean-of-2-max_ with CV events was stronger among the groups with a lower estimated basal CV risk: women, younger subjects, residents in southern Europe, subjects without prevalence of diabetes mellitus, and subjects with a lower basal SCORE2/SCORE2-OP. However, the disparity between groups varied according to the ultrasonographic variable considered, being greater for IMT_mean–max_ (with sporadically significant group × ultrasonographic variable interactions) and less marked for 1CC-IMT_mean-of-2-max_ and BIF-IMT_mean-of-2-max_.

As a simple practical example of how 1CC-IMT_mean-of-2-max_ may be employed in refining risk estimation, we computed the product of SCORE2/SCORE2-OP with 1CC-IMT_mean-of-2-max_ and with IMT_mean–max_ (Table 4).

Using the median of baseline SCORE2/SCORE2-OP (i.e., 8.25) as a cutoff, the sensitivity and specificity for the combined event were 70.1% and 54.8%, respectively. When SCORE2/SCORE2-OP was corrected using the IMT values, by simply computing a new variable consisting of the product of SCORE2/SCORE2-OP and IMT variables, the sensitivity was modestly reduced but the specificity was markedly increased (Table 4).

### 3.5. Calibration and Reproducibility

Calibration plots for 1CC-IMT_mean-of-2-max_ and BIF-IMT_mean-of-2-max_ are presented in Appendix A. According to the Hosmer–Lemeshow test, the 1CC-IMT_mean-of-2-max_ was well calibrated (Panel A), while the calibration for BIF-IMT_mean-of-2-max_ was less good (Panel B), although the *p*-value of the test (0.09) indicated that the deviation from a good fit did not reach significance.

Reproducibility measures for 1CC-IMT_mean-of-2-max_ and BIF-IMT_mean-of-2-max_ were obtained from duplicate scans of 159 subjects (125 intra-observer and 34 inter-observer). The intra-observer absolute differences (mean ± SD) between duplicate scans were 0.083 ± 0.095 for 1CC-IMT_mean-of-2-max_ and 0.184 ± 0.319 for BIF-IMT_mean-of-2-max_; the inter-observer absolute differences were 0.137 ± 0.179 and 0.195 ± 0.203, respectively. The intra-observer intraclass correlation coefficients were 0.89 and 0.94 for 1CC-IMT_mean-of-2-max_ and BIF-IMT_mean-of-2-max_, respectively; the respective inter-observer intraclass correlations coefficients were 0.74 and 0.78

### 3.6. Sensitivity Analyses

The exclusion of the two centers with a total follow-up ending at three years yielded very similar results in terms of HRs and AUC, indicating that a substantial bias due to disparity in follow-up times between centers is unlikely. In addition, we tested for potential biases due to the abnormally long follow-up in subjects from the Perugia center (up to 18 years). When their follow-up times were artificially censored at 12.5 years, again HRs and AUC were minimally affected.

If the predictive capacity was tested against the Framingham risk score, the AUC increase was still significant for 1CC-IMT_mean-of-2-max_ (+0.014, 95% CI 0.0002; 0.028, *p* = 0.04) and not significant for BIF-IMT_mean-of-2-max_ (+0.010, 95% CI −0.0024; 0.022, *p* = 0.12).

When we replaced the continuous variables IMT_mean–max_, 1CC-IMT_mean-of-2-max_, or BIF-IMT_mean-of-2-max_ with the dichotomous variables indicating the presence/absence of a plaque, the AUC increases, on top of SCORE2/SCORE2-OP, were consistently less significant: +0.025 (95% CI 0.000; 0.05, *p* = 0.05) for the presence of at least a plaque in the whole carotid tree, +0.003 (95% CI −0.006; 0.011, *p* = 0.52) for the presence of at least a plaque in left and right CC, and +0.009 (95% CI −0.008; 0.027, *p* = 0.29) for the presence of at least a plaque in the left and right Bif.

## 4. Discussion

The present study shows that a summary CIMT variable, i.e., 1CC-IMT_mean-of-2-max_, based on a simple measurement protocol, predicts future CV events with adequate sensitivity and specificity.

Other ultrasonographic variables, such as BIF-IMT_mean-of-2-max_, were significantly associated with CV events in Cox regression, but 1CC-IMT_mean-of-2-max_ was also able to significantly improve the ROC AUC when added to SCORE2/SCORE2-OP. Our group previously showed that more complex variables were able to improve the prediction of CV events over the Framingham risk score [7]. Among these, IMT_mean–max_, calculated as the average of the thickest values of eight segments of carotid arteries, was the one most strongly associated with the events, and the present study confirmed this result. However, despite its high predictive capacity, IMT_mean–max_ has some drawbacks: its calculation requires scanning the entire carotid tree, including the ICA, which is notoriously difficult to visualize and measure. Thus, it needs highly trained operators, and its protocol execution takes quite a long time (at least 20 min). It is likely that complex and demanding protocols, such as the one required for IMT_mean–max_, led to the idea, expressed in the ESC 2021 guidelines, that CIMT should not be recommended for risk assessment in combination with classic risk scores, primarily due to issues of standardization and the need for highly specialized operators [11]. In contrast, calculating the 1CC-IMT_mean-of-2-max_ does not require deep training, with much shorter execution times (3 times shorter than for IMT_mean–max_). It is sufficient to identify and measure the two maximum IMT values—one from the right carotid artery and one from the left—located at the first cm of the common carotid, a segment that is widely recognized as easy to visualize and measure.

Having identified a predictor using measures that are much easier to acquire has important clinical and epidemiological implications. First, its measurement protocol allows us to use 1CC-IMT_mean-of-2-max_ in mass screening and risk stratification in extensive primary prevention interventions, and for large epidemiological studies and clinical trials, in contrast with more demanding and time-consuming protocols, such as that required to measure IMT_mean–max_. As outlined, the measurement of 1CC-IMT_mean-of-2-max_ does not require (although it does not rule out) the presence of plaque, thus allowing continuous risk assessment in younger subjects without overt atherosclerosis. Of note, subgroup analysis indicated that the 1CC-IMT_mean-of-2-max_ was more strongly associated with events in lower-risk individuals and younger age groups, in line with findings from other studies using different CIMT measures [25]. Although we did not directly analyze participants under 54 years of age, a clear trend existed, suggesting that 1CC-IMT_mean-of-2-max_ could be particularly valuable for the portion of the population that could benefit most from CV risk estimation and prevention. Indeed, primary prevention is gaining importance and is recommended at an increasingly early age, considering that almost 50% of all CV events take place prior to reaching 65 years of age [26]. Enhanced efficacy in primary prevention during early adulthood might potentially alleviate persistent CV disease inequalities and decrease CV-disease-related mortality [27]. In contrast, CAC measurement has been found to be less predictive in younger individuals, where the high percentage of zero CAC values in low-risk subjects leads to a significant number of false negatives [28].

The absolute value of the increase in AUC when 1CC-IMT_mean-of-2-max_ is added to SCORE2/SCORE2-OP (1.7%) may be interpreted as negligible. However, a small increase in ROC AUC can result in a relevant improvement in predictivity. Our simulation showed that the addition of the variable “age” to a hypothetical prediction model including other five established risk factors yielded an AUC increase even lower than that observed with the addition of 1CC-IMT_mean-of-2-max_ to the SCORE2/SCORE2-OP. The same was true when we added the variables “sex” and “SBP” to a hypothetical model lacking them. Considering that age, sex, and SBP are among the most important determinants of CV risk, the increase in AUC associated with the results of the considered ultrasonographic variables is far from negligible.

An important issue in assessing the predictive ability of carotid ultrasonographic measures is the dualism of CIMT and plaque. For instance, the 2016 and 2021 ESC guidelines explicitly state that the use of CIMT in CV risk evaluation is not recommended, whereas the presence of plaques can be considered a reliable risk modifier. Clearly, the term CIMT can assume a dual interpretation, either in its etymological sense as a “measure of thickness” when referring to plaque definition, or as a “measure of wall hyperplasia and/or hypertrophy”, indicating an arterial wall modification other than atherosclerosis. Therefore, one of the main reasons for confusion in the literature, and which may have greatly contributed to the “lack of standardization” alleged in the ESC guidelines, is whether the measurement of CIMT should include plaques or not. For instance, Inaba et al. [29] report that in 27 out of 35 studies examined (77%), it was unclear whether plaques were included in CIMT measurements or not. The literature shows that CIMT including plaques has a higher predictive power than CIMT measured in PF areas (see, for instance, Inaba et al. [29]) and, in the present study, PF-CC-IMT_mean_ was the least performing ultrasonographic variable. In our group, CIMT has always (since 1986) been measured across different segments, regardless of the presence or absence of plaques. In this context, local maximal IMTs capture the thickest part of plaques, if present, and if no plaques are detected (or their thickness is below the specified threshold), they reflect any gradual thickening of the artery wall. We believe that the good predictive ability of the newly proposed variables stems from their capacity to assess both the size (and presence) of any plaques, as well as any minor changes in the wall thickness. The importance of focusing not only on plaque but also, in its absence, on vascular damage identified as increased intima-media thickness is well-documented by studies demonstrating that, even in the absence of plaques, the mere presence of increased intima-media thickness doubles the risk of vascular events [30]. On the other hand, our group has shown that the maximal detected IMT, a measure of the thickest lesion present in the carotid tree, and PF-CC-IMT_mean_ concurrently participate in the prediction of CV events [31]. In the present study, the variables indicating plaque presence/absence were less predictive than continuous variables measured in the same carotid areas. Indeed, from a purely statistical perspective, converting a continuous predictor into a dichotomous variable with a specific cutoff typically results in information loss, and this approach may have some advantages only if the relationship between the predictor and the risk of event is sharply discontinuous in the proximity of the cutoff. Instead, the relation of the HR for events was roughly linear with increasing values of 1CC-IMT_mean-of-2-max_, as it was with other continuous CIMT variables examined (Appendix A).

The significant predictive power demonstrated by 1CC-IMT_mean-of-2-max_ indicates that this variable properly accounts for the presence/absence and for the size of plaques in the common carotids; on the other hand, it confirms that wall thickness below the 1.5 mm cutoff still retains valuable predictive information for future events.

### Strengths and Limitations

The present study has some strengths and limitations.

Among the strengths are an extensive sample size exceeding 3000 individuals, recruited in different European countries, and well-balanced between women and men, and a lengthy follow-up, ensuring an adequate number of outcomes for analysis. In addition, the study benefits from a well-standardized ultrasound protocol, a centralized ultrasonographic measurement, and state-of-the-art analytical procedures.

This study has also some limitations. First, it is based on a single sample; thus, the present results need to be corroborated in further studies based on independent samples. Second, in this analysis, follow-up durations varied significantly across centers, which, in principle, could have introduced bias, although the Cox proportional hazard model is specifically designed to adequately deal with varying censoring times. Nevertheless, we performed a sensitivity analysis by excluding data from centers with very short follow-up periods or by truncating long follow-ups. In both instances, the results were only minimally affected. Third, our study includes individuals with at least three VRFs, recruited from five European countries, and within a limited age range (54–79 years); therefore, our findings should be extrapolated with caution to the general European population and to patients with fewer than three VRFs, in other age ranges, or from other continents. It is well established that IMT has a lower predictive capacity compared to plaque. Consequently, focusing on a segment with a low probability of plaque formation might be considered a limitation. However, since the new variables do not rely on the presence of plaques, and given that individuals without plaques but with an increased IMT still exhibit a twofold higher risk [30], we believe the proposed variable remains highly relevant. This is particularly true as it enables improved risk stratification in younger individuals or those at lower risk, where plaques are often uncommon or entirely absent. Fourth, the study is based on 2D ultrasonographic measurements, whereas more advanced technologies allow for 3D reconstruction of carotid plaques. Regarding the latter point, however, it should be considered that 3D imaging technologies require expensive equipment, intensive training, and longer scan times, and thus are not suitable for large-scale mass screening.

## 5. Conclusions

This study shows that a new protocol for the evaluation of 1CC-IMT_mean-of-2-max_, which is both easy to implement and to standardize, produces measures that can significantly improve the predictive ability of SCORE2/SCORE2-OP. Compared with more demanding protocols, the results of the simplified protocol are less performing, but the time and cost of execution make it “friendly” and suitable for extensive implementation in mass screening for risk assessment in primary prevention.

## Figures and Tables

**Figure 1 biomedicines-13-00584-f001:**
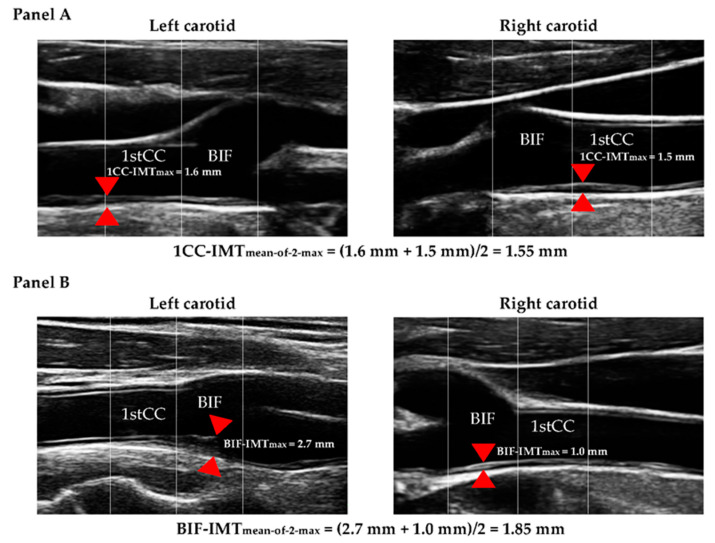
Didactic example of how 1CC-IMT_mean-of-2-max_ (**Panel A**) and BIF-IMT_mean-of-2-max_ (**Panel B**) are calculated. 1CC-IMT_mean-of-2-max_ is the average between the maximum IMT value identified in the 1st cm of the left and right common carotid artery. BIF-IMT_mean-of-2-max_ is the average between the maximum IMT value identified in the bifurcations of the left and right carotid arteries. The red triangle markers indicate the positions where the IMTmax measurements were taken. 1stCC: 1st cm of the common carotid artery; BIF: bifurcation; IMT: intima-media thickness. Definitions of 1CC-IMT_mean-of-2-max_ and BIF-IMT_mean-of-2-max_ are reported in Appendix A.

**Figure 2 biomedicines-13-00584-f002:**
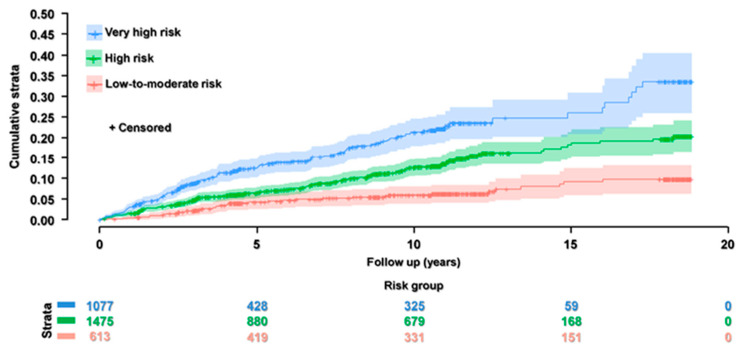
Kaplan–Meier curves for cumulative incidence stratified by SCORE2/SCORE2-OP class. Product limit Survival Estimates with the number of subjects at risk and 95% Hall–Wellner Bands. Logrank *p* < 0.0001.

**Table 1 biomedicines-13-00584-t001:** Baseline characteristics of the study participants without and with vascular events.

	Without Event (n = 2798)	With Event (n = 367)	*p*-Value
**Recruiting centre**			
Kuopio, n (%)	480 (17.2)	52 (14.2)	
Stockholm, n (%)	451 (16.1)	76 (20.7)	
Groningen, n (%)	470 (16.8)	48 (13.1)	0.03
Paris, n (%)	441 (15.8)	55 (15.0)	
Milan, n (%)	492 (17.6)	59 (16.1)	
Perugia, n (%)	464 (16.6)	77 (21.0)	
**Anthropometric variables**			
Male sex, n (%)	1237 (44.2)	213 (58.0)	<0.001
Age (years)	64.1 ± 5.4	65.0 ± 5.6	0.005
BMI (kg/m^2^)	27.2 ± 4.3	27.4 ± 4.3	0.37
Waist/hip ratio	0.92 ± 0.09	0.93 ± 0.08	<0.001
Diastolic blood pressure (mmHg)	81.9 ± 9.9	82.1 ± 10.1	0.92
Systolic blood pressure (mmHg)	141.3 ± 18.6	143.2 ± 19.0	0.13
**Smoking habits**			
Current smokers, n (%)	369 (13.2)	79 (21.5)	
Former smokers, n (%)	1038 (37.1)	147 (40.1)	<0.001
Never smokers, n (%)	1391 (49.7)	141 (38.4)	
Pack-years *	0.5 (0.0; 17.3)	8.3 (0.0; 26.3)	<0.001
**Biochemical markers**			
Total cholesterol (mg/dL)	215 ± 44	216 ± 42	0.58
HDL cholesterol (mg/dL)	49 ± 14	46 ± 12	<0.001
Triglycerides (mg/dL)	117 (84; 170)	123 (90; 182)	0.03
LDL cholesterol (mg/dL)	139 ± 39	141 ± 38	0.31
Uric acid (µmol/L)	5.3 ± 1.2	5.5 ± 1.2	0.002
hs-CRP (mg/L)	1.94 (0.82; 3.65)	2.15 (0.88; 4.29)	0.04
Blood glucose (mmol/L)	5.9 ± 1.7	6.0 ± 1.6	0.22
Creatinine (µmol/L)	80 ± 17	85 ± 20	<0.001
**Personal history (P.H.)**			
P.H. of hypertension, n (%)	1950 (69.7)	275 (74.9)	0.04
P.H. of diabetes, n (%)	729 (26.1)	104 (28.3)	0.35
SCORE2/SCORE2-OP	8.2 (5.6; 12.0)	9.9 (7.1; 14.5)	<0.001
**SCORE2/SCORE2-OP strata**			
Low-to-moderate risk, n (%)	575 (20.5)	38 (10.4)	
High risk, n (%)	1311 (46.8)	164 (44.7)	<0.001
Very high risk, n (%)	912 (32.6)	165 (45.0)	
**Family history (F.H.)**			
F.H. of CHD, n (%)	1628 (58.2)	227 (61.9)	0.16
F.H. of CVD, n (%)	991 (35.4)	132 (36.0)	0.84
F.H. of PVD, n (%)	286 (10.2)	41 (11.2)	0.57
**Therapies**			
Statins, n (%)	1087 (38.8)	132 (36)	0.29
Fibrates, n (%)	249 (8.9)	29 (7.9)	0.53
Fish oil, n (%)	107 (3.8)	18 (4.9)	0.32
Other lipid-lowering drugs, n (%)	19 (0.68)	3 (0.82)	0.73
Beta blockers, n (%)	615 (22.0)	88 (24.0)	0.39
Calcium antagonists, n (%)	437 (15.6)	60 (16.3)	0.72
ACE inhibitors, n (%)	520 (18.6)	81 (22.1)	0.11
Sartans, n (%)	388 (13.9)	56 (15.3)	0.47
Diuretics, n (%)	683 (24.4)	92 (25.1)	0.78
Anti-platelet agents, n (%)	385 (13.8)	87 (23.7)	<0.001
Insulin, n (%)	105 (3.8)	17 (4.6)	0.41
Other blood glucose lowering drugs, n (%)	1362 (48.7)	169 (46.1)	0.34
Estrogen supplementation, n (%)	157 (5.6)	12 (3.3)	0.06
**Ultrasonographic variables (mm)**			
1CC-IMT_mean-of-2-max_	1.00 ± 0.2	1.07 ± 0.2	<0.001
BIF-IMT_mean-of-2-max_	1.38 ± 0.5	1.52 ± 0.6	<0.001
PF-CC-IMT_mean_	0.68 ± 0.1	0.70 ± 0.1	<0.001
IMT_mean–max_	1.22 ± 0.3	1.34 ± 0.3	<0.001
**Presence of plaques**			
Presence of at least one plaque	1829 (65.4)	298 (81.2)	<0.001
Presence of at least one plaque in first cm CC	299 (10.7)	71 (19.3)	<0.001
Presence of at least one plaque in Bif	1588 (56.8)	257 (70.0)	<0.001

Values are n (%), mean ± SD, or median (interquartile range). *p*-values were calculated via the Wilcoxon test or via chi-square as appropriate. * Calculated excluding non-smokers. BMI: body mass index; HDL: high-density lipoprotein; LDL: low-density lipoprotein; hs-CRP: high-sensitivity C-reactive protein; SCORE2/SCORE2-OP: Systemic Coronary Risk Estimation-2/SCORE2-Older Persons; CHD: coronary heart disease; CVD: cerebrovascular disease; PVD: peripheral vascular disease; ACE: angiotensin-converting enzyme; CC: common carotid; 1CC: first cm of the common carotid; Bif: bifurcation; PF: plaque-free. Definitions of the ultrasonographic variables are reported in Appendix A.

**Table 2 biomedicines-13-00584-t002:** Time-dependent ROC analysis for ultrasonographic variable and for SCORE2/SCORE2-OP for prediction of vascular events.

	AUC for US Variable	AUC for SCORE #	Total AUC *	AUC Increase (95% CI)	*p*-Value
1CC-IMT_mean-of-2-max_	0.613		0.662	0.017 (0.003; 0.031)	0.014
BIF-IMT_mean-of-2-max_	0.592		0.654	0.009 (−0.007; 0.025)	0.24
		0.645			
IMT_mean–max_	0.643		0.672	0.029 (0.009; 0.049)	0.004
PF-CC-IMT_mean_	0.600		0.655	0.007 (−0.008; 0.021)	0.36

# Score: SCORE2/SCORE2-OP. * Total AUC: ROC AUC including both ultrasonographic variables and SCORE2/SCORE2-OP. ROC: Receiver Operating Characteristics; AUC: Area Under the Curve; US: ultrasonographic; CI: Confidence Interval; 1CC: first cm of the common carotid; BIF: bifurcation; IMT: intima-media thickness; PF: plaque-free. All ROC AUCs were estimated at 10 years of follow-up.

**Table 3 biomedicines-13-00584-t003:** Standardized HRs for the combined endpoint associated with different ultrasonographic variables.

US Variable	Standardized HR (95% CI)
1CC-IMT_mean-of-2-max_	1.20 (1.11; 1.30)
BIF-IMT_mean-of-2-max_	1.21 (1.10; 1.33)
IMT_mean–max_	1.35 (1.22; 1.48)
PF-CC-IMT_mean_	1.27 (1.14; 1.41)

US: ultrasonographic; HR: hazard ratio; CI: Confidence Interval; 1CC: first cm of the common carotid; BIF: bifurcation; IMT: intima-media thickness; PF: plaque-free. The analysis was stratified by center and the HRs were adjusted for SCORE2/SCORE2-OP.

**Table 4 biomedicines-13-00584-t004:** Sensitivity and specificity for the combined event using SCORE2/SCORE2-OP alone or multiplied (×) by ultrasonographic measures. The resulting variables were categorized using the respective medians as cutoffs.

	Cutoff	Sensitivity (%)	Specificity (%)
SCORE # (Alone)	8.25	70.1	54.8
SCORE # × 1CC-IMT_mean-of-2-max_	8.28	68.2	61.2
SCORE # × IMT_mean–max_	10.05	69.8	62.1

The “×” is the mathematical symbol for multiplication. # Score: SCORE2/SCORE2-OP; 1CC-IMT_mean-of-2-max_: average of two maximal CIMT measures taken, from a single angle, in the right and left common carotids; IMT_mean–max_: average of the maximal IMT measures obtained in eight segments of both carotid arteries.

## Data Availability

The original contributions presented in this study are included in the article/Appendix A. Further inquiries can be directed to the corresponding author.

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
