# Peer review of "Enhancing Cardiovascular Risk Prediction with a Simplified Carotid IMT Protocol: Evidence from the IMPROVE Study"

_biomedicines, 2025, doi:10.3390/biomedicines13030584_

Round 1

Reviewer 1 Report

Comments and Suggestions for Authors

The manuscript investigates the predictive utility of simplified carotid intima-media thickness (CIMT) measures for cardiovascular risk assessment. The study is based on a robust dataset from the IMPROVE cohort and introduces new CIMT-derived variables to streamline prediction. Overall, the manuscript adequately addresses, with a well-structured methodology, an issue that is not particularly relevant in clinical practice. 

The study tries to simplify CIMT measurements for widespread clinical use, addressing barriers like time and operator expertise. A cohort of 3,165 subjects with a median follow-up of 7.1 years ensures statistical power, but, considering the type of pathology and the numerous variables, this cohort could increase, perhaps involving numerous vascular surgery operating units that follow more specialists and practices these patients. 

The focus on new variables (1CC-IMTmean-of-2-max and BIF-IMTmean-of-2-max) and their comparison with existing measures is innovative.

The manuscript uses AUC improvements as a primary indicator of predictive power. While valid, the absolute increase (0.017 for 1CC-IMTmean-of-2-max) is marginal and could be contextualized more effectively. The use of standardized HRs is appropriate, but it would be valuable to include calibration plots for risk prediction models, as they provide a direct assessment of the agreement between predicted and observed risks.

The dataset excludes certain centers and relies on a restricted age range (54–79 years). These factors limit generalizability. Addressing this in the limitations section is essential, with an emphasis on how findings apply to younger or lower-risk populations.

While the protocol is described as "easy to standardize," practical details (inter-operator variability, training requirements, equipment dependency) are not fully addressed. Carotid IMT should in fact be evaluated by vascular surgeons with diagnostic expertise. Including these details would strengthen the argument for clinical adoption.

The manuscript contrasts its findings with current ESC guidelines but does not fully explain the discrepancy. For example, the guidelines emphasize the role of plaques over CIMT. The discussion would benefit from a deeper exploration of how 1CC-IMTmean-of-2-max reconciles with these recommendations.

Other suggestions for improvement: Strengthen the discussion around clinical implications and potential integration into existing risk stratification frameworks; include additional statistical validation methods, such as net benefit analysis or external validation of the models; address potential biases in the study design, such as those introduced by different follow-up times across centers.

Reviewer 2 Report

Comments and Suggestions for Authors

Dear authors, I enjoyed reading your carefully prepared article. I have given you some criticisms and suggestions below.

1. I recommend that you change the title of your article to fully reflect the content of the study. It would be useful to specify in the title what you mean by the expression "predictive ability of carotid IMT".

2. Abstract: Background/Objectives: You can move the statement that you use the "IMPROVE study database" in your study to the methods section of the abstract

3. Keywords: You can increase the keywords a few more. For example, you can add "intima-media thickness".

4. Introduction: The introduction of the study seems short. Therefore, I would like to suggest a further expansion. For example, other parameters used in predicting cardiovascular risk, if any, other than intima-media thickness increase, can be mentioned. You can also refer to the IMPROVE study.

5. Materials and methods: I recommend that a carotid ultrasonography image showing the measurement locations of carotid intima-media thickness be added to the article, if possible. This will make it easier for readers to visualize the measurement method.

6. Materials and methods: SCORE2/SCORE2-OP2 was mentioned in the statistical analysis section, but I think these scores should be mentioned in the materials and methods section beforehand.

7. I recommend that the long forms of the abbreviations ROC and AUC be used wherever they are first used, both in the abstract and in the main text.

8. Discussion: The article has a total of 23 references, and I consider this to be few. It seems that there are few references, especially in the discussion section. I think a more comprehensive comparison can be made with the data in the literature.

I hope these suggestions and critiques help make your article stronger.
